# Response of Astrocyte Subpopulations Following Spinal Cord Injury

**DOI:** 10.3390/cells11040721

**Published:** 2022-02-18

**Authors:** R. Vivian Allahyari, Nicolette M. Heinsinger, Daniel Hwang, David A. Jaffe, Javad Rasouli, Stephanie Shiers, Samantha J. Thomas, Theodore J. Price, Abdolmohamad Rostami, Angelo C. Lepore

**Affiliations:** 1Department of Neuroscience, Vickie and Jack Farber Institute for Neuroscience, Sidney Kimmel Medical College, Thomas Jefferson University, Philadelphia, PA 19107, USA; allahyari.vivian@gmail.com (R.V.A.); nheinsinger@gmail.com (N.M.H.); david.jaffe@students.jefferson.edu (D.A.J.); samantha.thomas@students.jefferson.edu (S.J.T.); 2Department of Neurology, Vickie and Jack Farber Institute for Neuroscience, Sidney Kimmel Medical College, Thomas Jefferson University, Philadelphia, PA 19107, USA; daniel.hwang@students.jefferson.edu (D.H.); javad.rasouli@students.jefferson.edu (J.R.); a.m.rostami@jefferson.edu (A.R.); 3Center for Advanced Pain Studies, Department of Neuroscience, University of Texas at Dallas, Richardson, TX 75080, USA; sis150030@utdallas.edu (S.S.); theodore.price@utdallas.edu (T.J.P.)

**Keywords:** astrocyte, SCI, spinal cord injury, synaptogenesis

## Abstract

There is growing appreciation for astrocyte heterogeneity both across and within central nervous system (CNS) regions, as well as between intact and diseased states. Recent work identified multiple astrocyte subpopulations in mature brain. Interestingly, one subpopulation (Population C) was shown to possess significantly enhanced synaptogenic properties in vitro, as compared with other astrocyte subpopulations of adult cortex and spinal cord. Following spinal cord injury (SCI), damaged neurons lose synaptic connections with neuronal partners, resulting in persistent functional loss. We determined whether SCI induces an enhanced synaptomodulatory astrocyte phenotype by shifting toward a greater proportion of Population C cells and/or increasing expression of relevant synapse formation-associated genes within one or more astrocyte subpopulations. Using flow cytometry and RNAscope in situ hybridization, we found that astrocyte subpopulation distribution in the spinal cord did not change to a selectively synaptogenic phenotype following mouse cervical hemisection-type SCI. We also found that spinal cord astrocytes expressed synapse formation-associated genes to a similar degree across subpopulations, as well as in an unchanged manner between uninjured and SCI conditions. Finally, we confirmed these astrocyte subpopulations are also present in the human spinal cord in a similar distribution as mouse, suggesting possible conservation of spinal cord astrocyte heterogeneity across species.

## 1. Introduction

Cellular heterogeneity within the central nervous system (CNS) has long been recognized since the classical descriptions of Santiago Ramon y Cajal [1]. Much of the focus on this heterogeneity has been on morphological, gene expression and functional diversity across neuronal populations, with dramatically less emphasis placed on glia. Astrocytes, a major glial subtype, perform a wide variety of functions both in the intact and diseased CNS. These functions include promoting synapse formation during development, extracellular matrix molecule production, regulation of blood–brain barrier maintenance and function and modulatory actions as part of the tripartite synapse such as extracellular glutamate uptake [2]. Several different molecularly distinct astrocyte subpopulations have been identified to date, and a number of groups have used various approaches to classify subpopulations of astrocytes [3,4,5,6,7]. Whether distinct populations of astrocytes are responsible for specific subsets of functions in vivo is currently being explored [8,9,10]. Additionally, how these distinct astrocyte populations respond to a CNS insult such as spinal cord injury (SCI) has not been determined.

Using a large screen of over 80 cell surface markers, the Deneen group recently classified subpopulations of astrocytes in multiple regions of the mouse CNS based on combinatorial expression of CD51, CD71 and CD63 [5]. Selection criteria required that the antibodies used for subpopulation classification mark 10–50% of all the astrocytes isolated from CNS tissue by fluorescence-activated cell sorting (FACS), reasoning that marking greater than 50% would represent ubiquitous astrocyte markers, while marking less than 10% would not be a high enough portion to reveal subpopulations. They identified five subpopulations of astrocytes, named Population A, B, C, D and E, that may serve different functions within the CNS. Interestingly, one subpopulation of mature adult astrocytes (Population C) was found to express greater levels of genes encoding for synapse-associated proteins and promoted the generation of more functional synapses when co-cultured with neurons, as compared both with other astrocyte subpopulations and with bulk unseparated astrocytes [5]. These findings suggest that this astrocyte subpopulation, which is found in relatively large numbers in the cerebral cortex where continuous remodeling of synapses occurs into adulthood, is selectively synaptogenic in vitro [5]. Moreover, Population C astrocytes appear to be largely absent from the mature intact adult spinal cord [5], suggesting that spinal cord astrocytes support synaptogenesis into adulthood at lower levels compared to other CNS locations. However, it remains unknown whether the synaptogenic potential of spinal cord astrocytes could change following CNS injury—such as via a shift in the distribution of these various subpopulations within the damaged spinal cord.

Following SCI, damaged neurons lose synaptic connections with pre-synaptic and post-synaptic neuronal partners, resulting in persistent loss of function and the therapeutic need for reconnection of this compromised neural circuitry [11,12,13,14]. SCI can also induce changes in other CNS cell populations, including glial cells such as astrocytes [15,16,17]. After trauma, astrocyte morphology and gene expression can be significantly impacted, and reactive astrocytes can take on new functional roles in injured tissue [2,18,19]. For example, astrocytes, and in particular scar-forming astrocytes surrounding a lesion, are critical for supporting axon regrowth following SCI; therapeutically stimulated axon growth is halted in the absence of these scar-forming astrocytes [18]. As astrocytes are known to change gene expression and function after CNS trauma, we hypothesized that spinal cord astrocytes may be able to acquire an enhanced synaptomodulatory phenotype in response to SCI by, for example, shifting toward a greater proportion of Population C cells and/or increasing expression of relevant synapse formation-associated genes within one or more of these astrocyte subpopulations.

Gene expression changes in bulk astrocytes from the spinal cord have previously been examined two weeks following SCI [18]; however, single-cell resolution of individual genes and in particular whether they are selectively expressed in specific subpopulations of astrocytes have not yet been examined. Furthermore, it remains unknown whether such gene expression changes last into the important chronic phase after injury. Here, we used a combination of flow cytometry and RNAscope in situ hybridization to determine whether subpopulations of spinal cord astrocytes shift in their distribution following a C2 hemisection (C2Hx) type SCI and also whether specific subpopulations have altered expression of synapse formation-associated genes, including at a long-term time point after injury.

## 2. Materials and Methods

### 2.1. Animals

All experimental protocols were carried out in compliance with the *Guide for the Care and Use of Laboratory Animals* (NIH, Bethesda, MD, USA), the Animal Research: Reporting of In Vivo Experiments (ARRIVE) guidelines, and the Thomas Jefferson University Institutional Animal Care and Use Committee (IACUC). Male and female adult mice between 3 and 9 months of age were used for this study. The following transgenic mouse lines were used: *Aldh1l1^CreER/+^* (The Jackson Laboratory, Stock #029655) and *Rosa26^tdTom/tdTom^* (Ai14) [20,21]. To identify subpopulations of astrocytes, 12–16 week old *Aldh1l1^CreER/+^;Rosa26^tdTom/tdTom^* mice were given 250 mg/kg tamoxifen via oral gavage for 3 consecutive days 2 weeks prior to C2 hemisection (C2Hx) or laminectomy-only surgery.

### 2.2. Spinal Cord Hemisection

The surgical procedure used for hemisection in mouse was based off previously established procedures conducted in rat by our lab and others [11,12,13,14]. Briefly, mice were anesthetized with 1.5–2% isoflurane in oxygen. Prior to first incision, animals received buprenorphine analgesic subcutaneously at 0.03 mg/kg. A bilateral laminectomy and durotomy was performed at the C2 spinal cord level. A lateralized section of the right half of the spinal cord was performed at C2, just caudal to the C2 dorsal rootlets, using a modified 30½ gauge needle (BD 305106) bent to a 90-degree angle (C2Hx). This section was repeated three times to ensure full severing of the right half of the spinal cord. The overlying muscles were then sutured in layers, and the skin closed using wound clips. Laminectomy-only controls (Lam) underwent the same surgical procedure, omitting the spinal hemisection and durotomy. Animals were then allowed to recover for 6 weeks following surgery, totaling at least 8 weeks between the last tamoxifen dose and tissue collection.

### 2.3. Dissociation for Flow Cytometry

Animals were briefly perfused with ice cold 1x PBS to remove blood from the CNS tissues. The cervical spinal cord was freshly dissected, and first coarsely chopped into smaller pieces by surgical scissors, followed by enzymatic dissociation with Liberase TL (Millipore Sigma, Burlington, MA, USA, Cat: 5401020001) dissolved in DMEM+F12 media at 37 °C in an incubator. A P1000 pipette tip was used to triturate the tissue at least five times at the end of the dissociation procedure. DMEM+F12 with 10% FBS was added at the end of the dissociation procedure to the dissociated cells to neutralize the enzyme, and the cell suspension was poured over a 100 μm cell strainer. Cells were pelleted and washed with PBS twice before being spun down in a 35% Percoll (Cytiva, Marlborough, MA, USA, Cat: 17089101) solution to separate out myelin and debris from isolated cells. Isolated cells were then washed with PBS twice before being stained with fluorochrome-conjugated primary antibodies at 4 °C for 20 min in 100 µL of FACS buffer (3% FBS in PBS). Cells were then washed once before analysis on a FACS Aria Fusion flow cytometer (BD Biosciences, San Jose, CA, USA). The following antibodies were used: FITC-rat anti-mouse CD71 (C2; BD Biosciences, Cat: 553266); BV650-rat anti-mouse CD51 (RMV-7, BD Biosciences, Cat: 740546); and APC-rat anti-mouse CD63 (NVG-2, eBioscience, Invitrogen, Waltham, MA, USA, Cat: 17-0631-80).

### 2.4. Human Tissue

Human spinal cord samples were recovered from organ donors at Southwest Transplant Alliance in Dallas, TX. The recovery of these tissues was approved by the Institutional Review Board of the University of Texas at Dallas, protocol number 15-237.

### 2.5. HiPlex RNAscope

A separate cohort of animals underwent C2Hx or Lam surgeries for evaluation of tissue using RNAscope in situ hybridization (ISH). Following perfusion with ice cold PBS, the cervical spinal cord was freshly dissected and flash frozen for ISH. RNAscope was performed on 30 µm-thick cryosectioned fresh frozen mouse spinal cord sections using the RNAscope HiPlex12 Reagents Kit (Advanced Cell Diagnostics, Newark, CA, USA), according to the manufacturer’s protocols. The following RNAscope HiPlex probes were used: Mm-Aldh1l1-T1 (Aldh1l1), Mm-Itgav-T2 (CD51), Mm-Cd63-T3 (CD63), Mm-Tfrc-T5 (CD71), Mm-Nrxn1-T7 (Nrxn1), Mm-Nrxn2-T6 (Nrxn2), Mm-Nlgn2-T4 (Nlgn2), Mm-Cdh2-T8 (Cdh2), Hs-TFRC-T6, Hs-ITGAV-T3, Hs-ALDH1L1-T2, Hs-CD63-T5. All tissues were first run with positive and negative control probes (species-specific) to ensure tissue and RNA quality for RNAscope ISH.

### 2.6. Microscopy

After each round of HiPlex RNAscope, images were obtained on an inverted confocal microscope (Leica-Sp8) using LASX software. Single-cell analysis of HiPlex probed ISH tissues from confocal z-stacks were collected with 40× oil objective, with a 0.5 μm step size. Images for each animal or donor were collected from at least 2 sections of tissue at each distance in the dorsal horn and ventral horn of the spinal cord. A total of 10 optical planes were collected in z-stacks, and collapsed for image analysis, ensuring that multiple cells in z-plane did not overlap. Cells were analyzed in ImageJ using code to identify the number of puncta present in each DAPI region of interest (ROI). After each round of imaging for HiPlex RNAscope, coverslips were removed according to the manufacturer’s protocol, fluorophores were cleaved, and new fluorophores were added (Advanced Cell Diagnostics, Newark, CA, USA). Immediately following this, we repeated imaging of the same cells on the same tissue slices using the additional ISH probes. We performed this for a maximum of three rounds of RNAscope in our study.

### 2.7. Quantification of RNAscope Puncta

Individual mRNA puncta were counted in each DAPI region of interest (ROI) using a custom Image J macro in which ROIs were determined based on DAPI signal, and cutoffs for number of puncta/transcripts required to be a “positive” cell for each gene are as follows (for the mouse tissue study): *Itgav*: 8; *Aldh1l1*: 3; *CD63*: 5; *Tfrc*: 3; *nlgn2*: 1, *nrxn1*: 1; *nrxn2:* 1; *cdh2*: 1. Cutoffs were identified based on manual single cell analysis over multiple images to determine positive vs. negative cells. Human tissue cutoffs were as follows: *Itgav*: 8; *Aldh1l1*: 3; *CD63*: 5; *Tfrc*: 3.

### 2.8. Statistical Analyses

Experiments included both male and female mice, and no differences between sexes were observed. All statistical analyses were performed using Prism 9 (GraphPad 6.01). Assignment of mice to groups, surgical procedures and analyses were all conducted in a blinded manner.

## 3. Results

### 3.1. Astrocyte Subpopulations Can Be Isolated from Adult Spinal Cord Using Flow Cytometry

Given the plasticity of astrocyte function and gene expression after SCI [2,19,22], we first examined whether there were differences in astrocyte subpopulation distribution between injured and uninjured animals. To do so, we isolated astrocytes from the spinal cord of adult mice that received either C2 hemisection (C2Hx) SCI or laminectomy-only (Lam), and performed flow cytometry to assess expression of the cell surface markers CD51, CD71 and CD63, categorizing astrocytes based on combinatorial expression of these three markers. We used a genetic inducible recombination strategy to permanently mark and identify *Aldh1l1*-expressing astrocytes, as *Aldh1l1* is a pan-astrocyte marker in the adult CNS [23]. Transgenic mice in which CreER is targeted to the *Aldh1l1* locus (*Aldh1l1^CreER/+^*) were crossed with the Rosa26 tdtomato (tdTom) reporter line (*Rosa26^flSTOPfltdtomato^*, line Ai14) to generate *Aldh1l1^CreER/+^;R26^tdT/tdT^* mice. Following tamoxifen administration, all astrocytes were permanently labeled with the red fluorescent tdTom reporter protein, allowing them to be sorted by subsequent flow cytometry to identify cells of interest.

Tamoxifen was administered via oral gavage to *Aldh1l1^CreER/+^;R26^tdT/tdT^* mice 2–4 weeks before either C2Hx or Lam control surgery to allow sufficient time for astrocytes to express the tdTom reporter. We chose to use a C2Hx model because a majority of SCI patients experience cervical trauma [24]. The cervical spinal cord, encompassing astrocytes both within and adjacent to the lesion, was dissected 6 weeks post-surgery. Cells were dissociated by mechanical force and enzymatic dissociation, and flow cytometry was used to classify tdTom^+^ astrocytes into the five subpopulations, as previously described [5]. These subpopulations each express different combinations of the markers CD51, CD71 and CD63. Population profiles of cell surface marker expression are shown in Figure 1a. We isolated astrocytes using flow cytometry (Figure 1b), and found that the vast majority of cervical spinal cord astrocytes were of Population A in both injured and uninjured spinal cords, while there were significantly fewer of the other subpopulations in the cervical spinal cord (Figure 1c–e), consistent with previous reporting of astrocytes from the entire spinal cord neuraxis [5]. We found a decrease in Population A after C2Hx, as compared to Lam controls (37.89% and 64.65%, respectively, Figure 1e). Furthermore, we found an increase in the percentage of Population E astrocytes after C2Hx compared to Lam (14.08% and 4.01%, respectively, Figure 1e). Interestingly, the subpopulation of astrocytes previously reported to differentially enhance synapse formation (Population C) remained unchanged between Lam and C2Hx and also represented only a small percentage of all tdTom^+^ astrocytes in both uninjured and SCI conditions (Figure 1e). These data suggest that C2Hx resulted in only a very modest change in astrocyte subpopulation distribution in the cervical spinal cord, and that previously reported synaptogenic astrocytes are present at low levels in both the injured or uninjured spinal cord.

### 3.2. Astrocyte Subpopulations Can Be Identified in Spinal Cord via RNAscope in Situ Hybridization

Traditional fluorescence immunohistochemistry and ISH protocols are limited in the number of antibodies and ISH probes that can be used simultaneously on the same section. To increase the number of probes we could employ in a single tissue sample, we used HiPlex RNAscope ISH to examine subpopulations of astrocytes in cervical spinal cord sections, which allows for simultaneous probing of up to 12 targets within the same single cells. HiPlex RNAscope utilizes unique probe tails which bind cleavable fluorophores, allowing for repeated imaging, cleaving of fluorophores, and addition of new fluorophores in the same section. Moreover, we examined whether there were spatially distinct, region-specific changes in astrocyte subpopulations by performing analyses separately in the dorsal horn, ventral horn and white matter of the spinal cord, as well as separately close to the lesion site and in the surrounding intact spinal cord. This approach provided dramatically increased spatial resolution compared to our flow cytometry assessment of all cervical spinal cord astrocytes (Figure 1). Furthermore, we used HiPlex RNAscope to quantify expression of various synapse formation-associated genes in the diverse astrocyte subpopulations in injured and uninjured cervical spinal cord tissue.

We performed C2Hx or C2 Lam-only surgeries and freshly dissected cervical spinal cord for histological analyses 6 weeks after surgery to match our flow cytometry experimental timeline. HiPlex RNAscope ISH was performed on tissue sections from the C2 level at the interface between the lesion site and intact tissue, as well as in caudal intact spinal cord at the C4 level (Figure 2a). This approach allowed for exploration of subpopulation changes at multiple locations relative to the SCI site, which is important given that the response of astrocytes can vary greatly with distance from an insult [2,25,26]. We probed for the following genes to examine subpopulations of astrocytes in C2Hx and Lam tissues: *Aldh1l1* for all astrocytes; *itgav* for CD51; *tfrc* for CD71; and *CD63* for CD63 (Figure 2b,c). Representative individual astrocytes from Populations A, C and E are shown in Figure 2c. With RNAscope ISH, individual puncta represent single mRNA transcripts; therefore, we quantified the number of transcripts present in each DAPI nucleus, and determined the percentage of astrocytes that corresponded to each subpopulation. We performed quantification of grey matter astrocyte subpopulations in both the dorsal and ventral horn. We found that there were no differences between C2Hx mice and Lam controls at the lesion site in the percentage of cells in each astrocyte subpopulation in either the dorsal or ventral horn (Figure 2d,e). Additionally, there were no differences between Lam and C2Hx in the percentage of astrocytes in each subpopulation at the intact caudal location, again in both the dorsal and ventral horn (Figure 2f,g). These data suggest that the distribution of astrocyte subpopulations was not different between SCI and Lam controls both near the lesion site and in intact tissue several segments away from the injury. In addition, there were no region-specific differences in astrocyte subpopulation distribution across gray matter locations, both in the intact spinal cord and after SCI. Furthermore, similar to our flow cytometry findings, we also found that the synaptogenic Type C astrocytes are found at very low levels: in both the intact and injured cervical spinal cord; at both dorsal horn and ventral horn locations; and in the SCI animals at both lesion site and caudal intact locations.

We also quantified total numbers of *Aldh1l1*-positive astrocytes at the C2 lesion site location in the RNAscope analysis, as Aldh1l1 is a pan-astrocyte marker. At 6 weeks post-surgery, there was a large increase in total astrocyte numbers in ventral horn of the SCI mice compared to uninjured controls (Lam: 19.4+/−2.0 *Aldh1l1*^+^/DAPI^+^ astrocytes per unit area; SCI: 36.5+/−4.0 *Aldh1l1*^+^/DAPI^+^ astrocytes; *p* = 0.003, t-test; *n* = 8 animals per group). These data demonstrate that, while we did not observe a change in distribution of the various astrocyte subpopulations (i.e., Type A–E) after SCI with RNAscope analysis, there was an expected astrogliosis response after SCI that persisted to the long-term 6 week time point.

### 3.3. Expression of Synapse Formation-Associated Genes in Spinal Cord Astrocytes

After SCI, synaptic connectivity is lost between injured neurons within damaged circuits. However, injury can induce spontaneous new synapse formation following this initial degeneration [27]. In some instances, spontaneous synapse formation after SCI can be adaptive; for example, the axons of spared neurons can sprout to generate novel circuit connections that lead to some degree of functional recovery [27]. Of note, however, spontaneous synaptogenesis after SCI could also lead to adverse consequences such as spasticity and chronic neuropathic pain [28,29,30]. Population C astrocytes have been previously reported to differentially enhance synaptogenesis between co-cultured neurons in vitro compared to other astrocyte subpopulations [5]. Moreover, the expression of synapse-associated genes was also reported to be enriched in Population C astrocytes, when compared with other subpopulations [5]. Although we show above that the percentage of Population C astrocytes was very low in both intact and injured spinal cord and also did not change after SCI, it remains unknown whether synapse-related gene expression was altered in each astrocyte subpopulation after injury. Therefore, we examined synapse formation-associated genes present in each astrocyte subpopulation in both the dorsal and ventral horn at 6 weeks after C2Hx or Lam using HiPlex RNAscope, including at the lesion site and in caudal intact spinal cord tissue.

For our analysis, we used the dataset from John Lin et al., 2017 to choose genes that are specifically enriched in Population C when compared to Population A and bulk astrocytes. Interestingly, several genes typically associated with neuronal synaptic connectivity were specifically enriched in Population C astrocytes: *neuroligin 2 (nlgn2)*, *neurexin 1 (nrxn1)*, *neurexin 2 (nrxn2)* and *cadherin 2 (cdh2)*. On the contrary, genes typically associated with astrocyte-mediated regulation of synaptic connectivity (i.e., thrombospondins, SPARC, Hevin, glypicans, etc. [31,32,33]) were not enriched in Population C astrocytes compared with Population A or bulk astrocytes derived from cerebral cortex [5]. We therefore probed for *nlgn2, nrxn1, nrxn2* and *cdh2* (in combination with *Aldh1l1*, *itgav*, *tfrc* and *CD63*) using HiPlex RNAscope ISH in the spinal cord 6 weeks after C2Hx or Lam (Figure 3). We found there were no significant differences in the percentage of total DAPI^+^ cells expressing *nlgn2*, *nrxn1*, *nrxn2* or *cdh2* in dorsal horn between C2Hx and Lam tissues, either at the lesion site or in caudal intact tissue (Figure 4a–d). Consistently, there were also no differences in percentage of total DAPI^+^ cells expressing *nlgn2*, *nrxn1*, *nrxn2* or *cdh2* in the ventral horn between C2Hx and Lam tissues, again at both the lesion site and in caudal intact tissue (Figure 4e–h).

We then specifically assessed percentage of total astrocytes (*Aldh1l1*^+^ cells) that expressed *nlgn2*, *nrxn1*, *nrxn2* or *cdh2* in the dorsal and ventral horn at the lesion site and in caudal intact tissue (Figure 4i–p). We found that almost all *Aldh1l1*^+^ astrocytes expressed *nlgn2*, *nrxn1*, *nrxn2* or *cdh2* in the dorsal horn in both injured and uninjured mice (Figure 4i–l). We also found that approximately 80% of all *Aldh1l1*^+^ astrocytes expressed *nlgn2* in the ventral horn (Figure 4m) and that roughly 60% of all *Aldh1l1*^+^ astrocytes expressed *nrxn2* in the ventral horn in both injured and uninjured tissue (Figure 4o). Additionally, nearly all *Aldh1l1*^+^ astrocytes in the ventral horn expressed *nrxn1* and *cdh2* in both C2Hx and Lam animals (Figure 4n,p). These data show that a large percentage of all astrocytes expressed synapse formation-associated genes in both the intact and injured spinal cord, and that there were some subtle region-specific differences in expression patterns for these genes in grey matter spinal cord astrocytes. In addition, SCI did not induce significant alterations in these synapse formation-associated gene expression profiles in spinal cord astrocytes.

We next quantified the percentage of each astrocyte subpopulation that expressed *nlgn2*, *nrxn1*, *nrxn2* or *cdh2* in the dorsal and ventral horn of the spinal cord (Figure 5). We found (1) that all of the examined synapse formation-associated genes were expressed in each of the subpopulations of astrocytes, (2) that there was not differential expression of these genes across astrocyte subpopulations, and (3) that there were no differences in expression observed between C2Hx and Lam controls in each of these astrocyte subpopulations.

In addition to the percentage of astrocytes in each subpopulation expressing these synapse formation-associated genes, we examined the total number of transcripts present per *Aldh1l1*^+^ astrocyte after C2Hx or Lam surgery at both the lesion site and in caudal intact tissues. We found that SCI did not induce any significant changes in the total number of transcripts of *nlgn2, nrxn1, nrxn2* or *cdh2* expressed in each *Aldh1l1*^+^ astrocyte of the dorsal horn between C2Hx mice and Lam controls (Figure 6a–d). Consistently, we also found that there were no significant differences between C2Hx and Lam controls in the number of transcripts per *Aldh1l1*^+^ astrocyte for *nlgn2*, *nrxn1*, *nrxn2* or *cdh2* in the ventral horn (Figure 6e–h). In the ventral horn, we also performed analysis of putative neurons, which we identified based on their very large *Aldh1l1^-^* DAPI^+^ nucleus. Although we observed that these ventral horn neurons expressed significantly greater numbers of synapse formation-associated gene transcripts (*nlgn2*, *nrxn1*, *nrxn2*, and *cdh2*) compared to ventral horn astrocytes, there were no significant differences noted in expression between C2Hx and Lam.

Collectively, these data demonstrate that all cervical spinal astrocyte subpopulations expressed similar levels of this panel of synapse formation-associated genes and that this expression profile did not change in response to SCI. Moreover, these data suggest that astrocytes expressed relatively lower levels of historically neuronal synapse formation-associated genes when compared with adjacent neurons present at the same anatomical location.

### 3.4. Astrocyte Subpopulation Characterization in Human Spinal Cord

Our experiments thus far have examined astrocytes subpopulations in the mouse spinal cord, yet it remains unknown whether the human spinal cord contains a similar astrocyte subpopulation distribution based on these cell surface marker criteria. Gaining a deeper understanding of human astrocyte subpopulations can lead to potential therapies specifically targeting individual cell populations. We therefore examined intact fresh-frozen human lumbar spinal cord tissue, and performed HiPlex RNAscope ISH for *Aldh1l1*, *itgav* (CD51), *tfrc* (CD71) and *CD63* (CD63) to evaluate subpopulations of astrocytes (Figure 7). We found that Populations A–E were all present in the dorsal and ventral horn of the adult human spinal cord (Figure 7c,d). Consistent with the mouse cervical spinal cord, the majority of human dorsal horn astrocytes were Population A (Figure 7c). In the ventral horn, Populations A and E were both in larger proportions than Populations B, C and D. Also consistent with the data from the mouse spinal cord, there was a notable relative absence of Population C astrocytes, suggesting that these putatively synaptogenic astrocytes were present at low levels in the human spinal cord. This is the first analysis of distinct astrocyte subpopulations in the intact human spinal cord. Our findings suggest that the presence of these astrocyte subpopulations and their relative distribution may be conserved across species.

## 4. Discussion

Astrocytes may be classified in a number of ways, including morphologically, spatially, molecularly and functionally [2,22,34]. We chose to use the Deneen group’s classification into subpopulations A-E because this previous work identified a functionally distinct subpopulation (Population C) that differentially supports synaptogenesis [5]. Interestingly, Population C astrocytes were found to be in large proportion in the cortex where continuous remodeling of synaptic circuits occurs into adulthood, but were found at much lower levels in the uninjured adult spinal cord [5], leading us to question whether astrocyte subpopulation distribution changes following insults such as SCI. In our study, we found that the distribution of these subpopulations was not altered 6 weeks following SCI, and we also demonstrated that none of these specific subpopulations exhibited differential gene expression changes in synapse formation-associated genes in response to injury. There is a possibility that astrocytes characterized into subpopulations by other classification schemes might yield different results, such that, for example, distinct populations may differentially increase or decrease expression of synapse-associated genes after damage. Along these lines, following CNS injury, a pan-astrocyte response in the expression of certain genes can occur, whereby individual subpopulations that are present in intact tissues all respond to an insult in a similar way [19,35]. For example, the vast majority of astrocytes near a lesion upregulate glial fibrillary acidic protein (GFAP) in response to mechanical injury in a distance dependent manner [25,35]. Although our data suggest that there was no difference in the gene expression levels of *neuroligin 2*, *neurexin 1*, *neurexin 2* or *cadherin 2* at 6 weeks after SCI in Populations A–E, it is possible that other astrocyte classification systems would yield alternative results for these and other synapse-related genes. Future analyses of various astrocyte subpopulations after SCI could elucidate such potential differences.

Astrocyte responses to CNS insults vary depending on the time point analyzed after injury. In the initial acute and subacute phases post-injury, the astrocyte response can differ in a number of ways compared to the chronic phase. Although we only analyzed a long-term time point in this study, it is possible that during the earlier phases after SCI there are significant alterations in astrocyte subpopulation distribution and/or in the expression of genes such that those involved in regulating synapse formation. Future analysis at various times during the acute and subacute period will be important for understanding the temporal evolution of the response of astrocyte subpopulations to SCI. Furthermore, it will be important to determine whether this response varies with distance from the lesion site and across various gray and white matter structures.

We found that astrocytes from the intact human spinal cord can also be classified using the Deneen group’s system into subpopulations A–E. Consistent with our study, human glioblastoma tissue also contains these astrocyte subpopulations [5]. Although we identified astrocytes of the same subpopulations in human spinal cord tissues based on histological analysis of differential expression of a cell surface marker panel, it still remains unclear whether these human spinal cord astrocytes possess the same functional characteristics (e.g., synaptogenic properties) in vitro and in vivo as the mouse experiments. Further studies using human induced pluripotent stem cells (iPSCs) differentiated to various astrocyte fates may represent one approach to shed light on this possibility if the same astrocyte subpopulations can be generated using iPSCs.

Following CNS injury, there is a multicellular response that results in damage to passing axons, as well as consequent synaptic disconnection of neural circuitry both close to the lesion and at distances far removed from the injury site [36]. A variety of molecular mechanisms limit the regrowth of both damaged axons and spared fibers, as well as the synaptic connectivity of these axon populations with post-synaptic targets [2,36]. After CNS trauma such as SCI, affected neural circuits can undergo a degree of plasticity/remodeling via processes such as axonal sprouting and synapse formation, which in some cases can promote modest functional recovery [36]. This suggests that enhancing synaptic connectivity after SCI is a potentially important target, and the role played by astrocytes in this plasticity process may be particularly important. The relative absence of synaptogenic Population C astrocytes in both the intact and injured spinal cord that we observed in this study, along with previous studies showing relatively little synaptic reconnection of pharmacologically-stimulated regenerating axons with their target neurons—particularly at chronic time points post-injury [12], suggests that the astrocyte subpopulation distribution present in the spinal cord after SCI (i.e., mostly Population A) does not robustly support synaptogenesis. The development of novel approaches to shift the distribution of the endogenous astrocyte makeup selectively toward a particular subpopulation (s) and/or the exogenous delivery of a particular subpopulation represent potentially promising therapeutic strategies. At the same time, maladaptive circuit plasticity events after SCI can drive the unwanted development of outcomes such as chronic pain, hyperreflexia and autonomic dysreflexia [37,38,39]. Therefore, it is critical for the field to first understand how various astrocyte subpopulations can impact specific adaptive and maladaptive plasticity processes before interventions are employed to alter the astrocyte landscape.

## 5. Conclusions

We used a combination of flow cytometry and RNAscope in situ hybridization to categorize astrocyte heterogeneity into previously described subpopulations at a single-cell level in uninjured mice and in response to SCI [5]. Following cervical SCI, astrocyte subpopulation distribution remained relatively unchanged compared to uninjured controls, including at multiple distances from the lesion and in various grey matter structures. We also found that synaptogenic Type C astrocytes are found at very low levels in both the intact and injured spinal cord. Additionally, we found that cervical spinal cord astrocytes expressed relatively similar levels of synapse formation-associated genes across all subpopulations and also between SCI and uninjured conditions. Finally, we showed that these astrocyte subpopulations can be found in a similar distribution in human spinal cord, suggesting possible conservation of astrocyte heterogeneity in the spinal cord across species. The field of astrocyte heterogeneity is growing at a rapid and exciting pace, resulting in the identification of astrocyte subpopulations within and across CNS regions and also generating new technologies for the study of these cells using both in vitro and in vivo experimental systems. Understanding diseased conditions such as SCI in the context of this diversity of astrocyte populations present in the intact and damaged CNS will be critical for the development of therapies across the spectrum of nervous system disorders.

## Figures and Tables

**Figure 1 cells-11-00721-f001:**
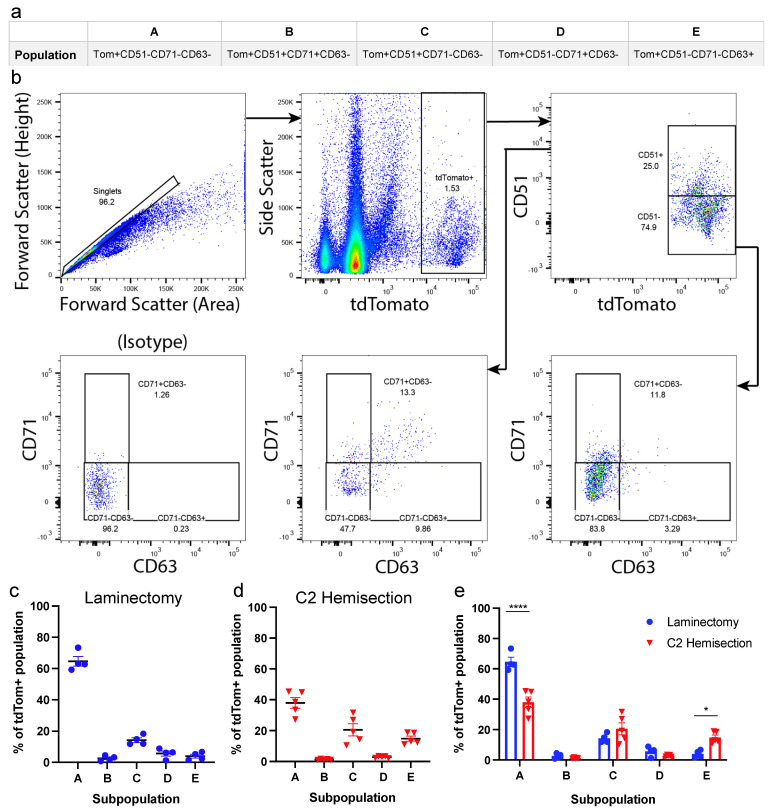
Identification of astrocyte subpopulations in the intact mouse spinal cord and after C2 hemisection SCI. (**a**) Cell population nomenclature (**A**–**E**) relating to cell surface marker combination. (**b**) Flow cytometry hierarchical gating (arrows) was used to identify subpopulations of astrocytes in Aldh1l1^CreER/+^;R26^tdTom/tdTom^ mice. (**c**–**e**) Flow cytometry analysis of astrocyte subpopulations in mice with laminectomy-only (Lam, c, *n* = 4 mice) and C2 Hemisection (C2Hx, d, *n* = 5 mice) surgeries. (**e**) Comparison of subpopulations of astrocytes from Lam and C2Hx mice 6 weeks post-SCI. Data points represent individual animals, error bars represent mean +/− SEM. Statistical significance was assessed by two-way ANOVA with Sidhak’s multiple comparisons. * *p* < 0.05, **** *p* < 0.0001.

**Figure 2 cells-11-00721-f002:**
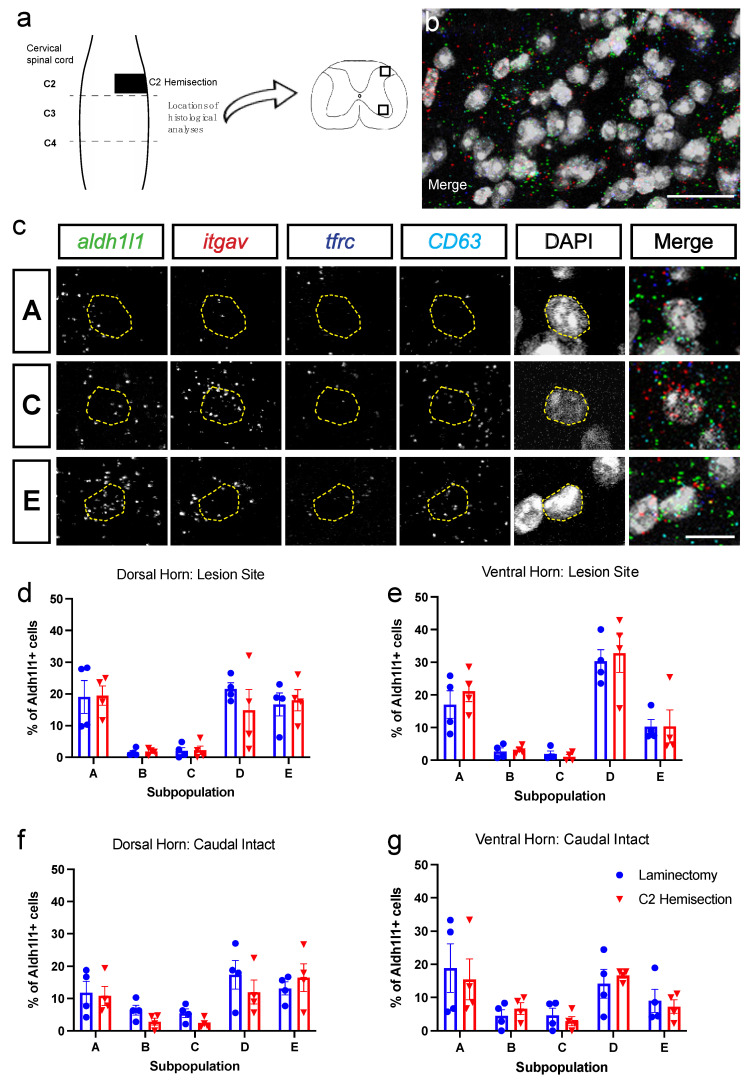
Subpopulations of astrocytes were unchanged after C2Hx injury when assessed by HiPlex RNAScope in situ hybridization. (**a**) Schematic of spinal cord demonstrating C2Hx location and sampling locations at lesion site (C2) and intact caudal intact cord (C4). Boxes in (**a**) depict location of tissue analysis. (**b**) Low magnification HiPlex RNAscope in situ hybridization probing for aldh1l1 (green), itgav (CD51, red), tfrc (CD71, blue, and CD63 (cyan). (**c**) Representative cells in Populations A, C and E. (**d**–**g**) Quantification of each subpopulation in C2Hx and Lam groups in dorsal horn (**d**,**f**) and ventral horn (**e,g**) of lesion site (**d,e**) and intact caudal spinal cord (**f,g**). Scale bars: 25 µm (**b**), 10 µm (**c**). *n* = 4 mice per group.

**Figure 3 cells-11-00721-f003:**
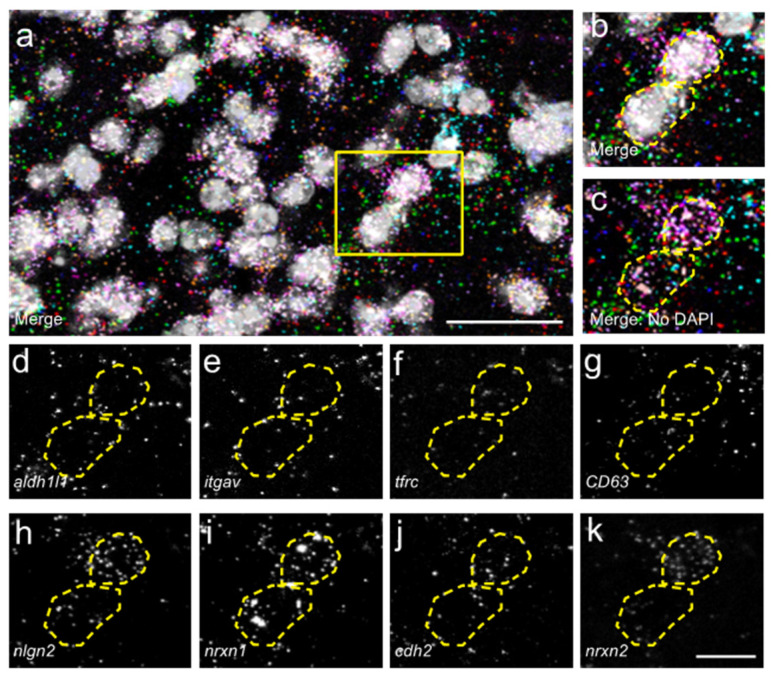
Expression of synapse formation-associated genes in spinal cord astrocytes. (**a**–**j**) Maximum projection images from confocal z-stacks of RNAscope in situ hybridization for astrocyte subpopulation defining cell surface markers (**d**–**g**) and synapse-associated genes (**h**–**k**). Merged image with DAPI counterstain shown in (**b**), and without DAPI counterstain (**c**). Scale bars, 25 µm (**a**), 10 µm (**b**–**k**).

**Figure 4 cells-11-00721-f004:**
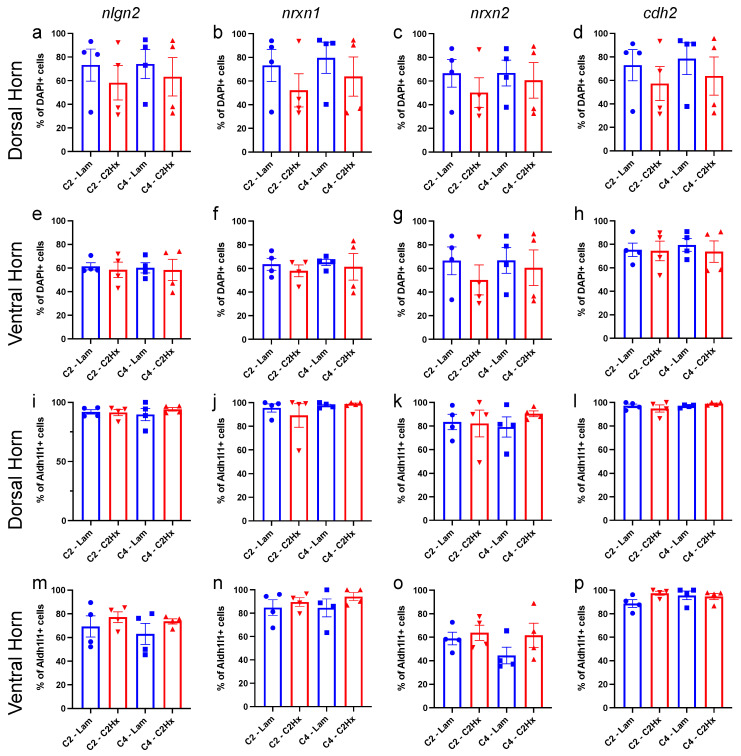
Synapse formation-associated genes were expressed in a large percentage of all astrocytes in the intact and injured spinal cord. (**a**–**p**) Quantification of the percentage of all nuclei (**a**–**h**) and all astrocytes (**i**–**p**) expressing genes for nlgn2 (**a,e,i,m**), nrxn1 (**b,f,j,n**), nrxn2 (**c,g,k,o**) and cdh2 (**d,h,l,p**) in dorsal horn (**a**–**d,i**–**l**) and ventral horn (**e**–**h,m**–**p**) of the spinal cord at the lesion site (C2-Lam, C2-C2Hx) and caudal to the lesion site (C4-Lam, C4-C2Hx). Data points represent individual animals. *n* = 4 animals per group. Error bars represent mean ± SEM. Statistical significance was assessed by one-way ANOVA. *n* = 4 mice per group.

**Figure 5 cells-11-00721-f005:**
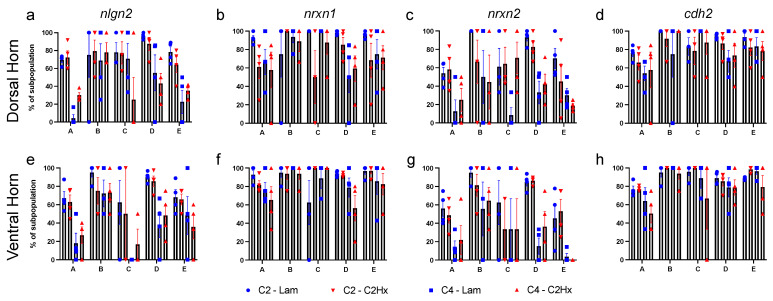
Astrocyte subpopulation expression of synapse formation-associated genes. (**a**–**h**) Quantification of the percentage of each astrocyte subpopulation expressing mRNA for nlgn2 (**a,e**), nrxn1 (**b,f**), nrxn2 (**c**,**g**) and cdh2 (**d,h**) in dorsal horn (**a**–**d**) and ventral horn (**e**–**h**) of the spinal cord at the lesion site (C2-Lam, C2-C2Hx) and caudal to the lesion site (C4-Lam, C4-C2Hx). Data points represent individual animals. *n* = 4 animals per group. Error bars represent mean ± SEM. *n* = 4 mice per group.

**Figure 6 cells-11-00721-f006:**
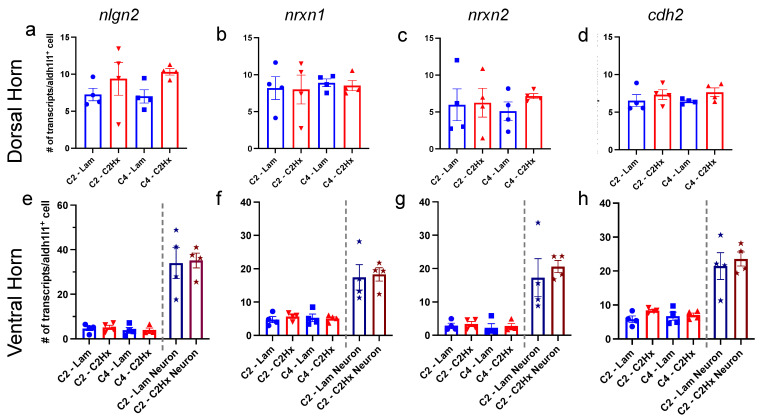
Astrocyte synapse formation-associated gene expression was unchanged after SCI. Quantification of the average number of transcripts of nlgn2 (**a,e**), nrxn1 (**b,f**), nrxn2 (**c,g**) and cdh2 (**d,h**) in each astrocyte or neuron (ventral horn only) in the dorsal horn (**a**–**d**) and ventral horn (**e**–**h**) of the spinal cord at the lesion site (C2-Lam, C2-C2Hx) and caudal intact tissue (C4-Lam, C4-C2Hx). Data points represent individual animals. *n* = 4 animals per group. Error bars represent mean ± SEM.

**Figure 7 cells-11-00721-f007:**
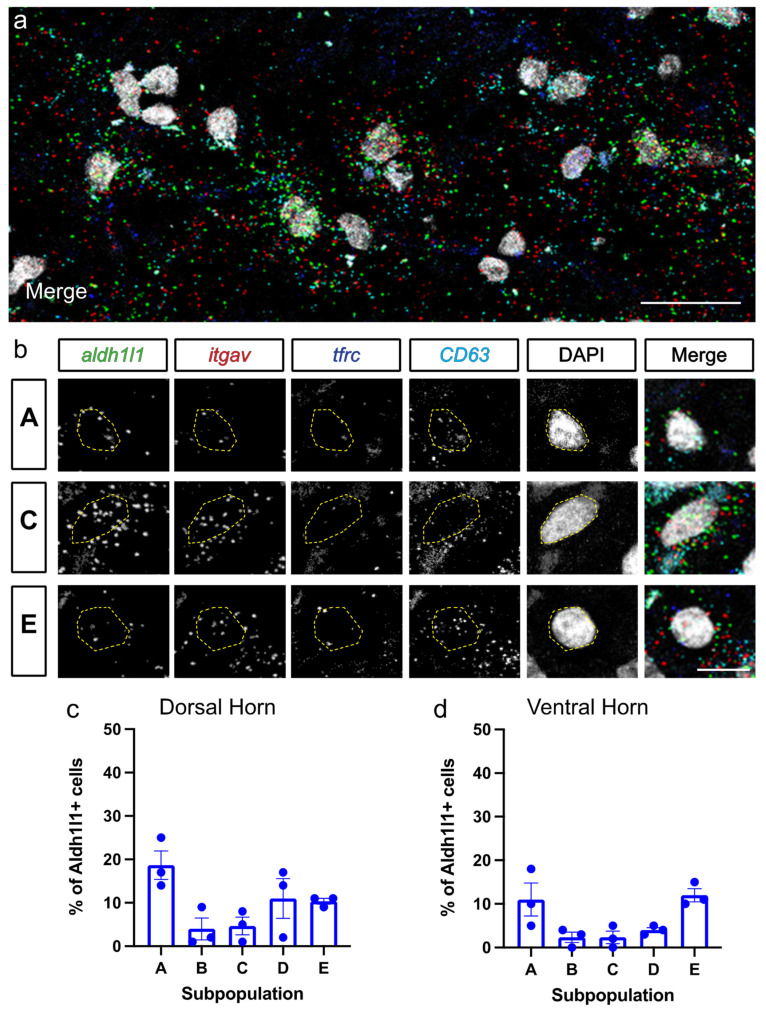
Astrocyte subpopulation characterization in human spinal cord. (**a**) Lower magnification image of RNAscope labeling of human spinal cord. (**b**) Representative cells in Populations A, C and E. (**c,d**) Quantification of the percentage of Aldh1l1+ astrocytes corresponding to each subpopulation in intact human lumbar spinal cord sections from dorsal horn (**b**) and ventral horn (**c**). Data points represent individual tissue donors, error bars represent mean +/− SEM. Scale bars, 25 µm (**a**), 10 µm (**b**).

## Data Availability

Not applicable.

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
