# Peer review of "Response of Astrocyte Subpopulations Following Spinal Cord Injury"

_cells, 2022, doi:10.3390/cells11040721_

Round 1

Reviewer 1 Report

This excellent manuscript deals with astrocyte heterogeneity in the intact and injured spinal cord. Particularly, they analyzed one astrocyte subpopulation, population C shown to have enhanced synaptogenic properties. Using flow cytometry and RNAscope in situ hybridization, the authors demonstrate that astrocyte subpopulation in the spinal cord did not change to a selectively synaptogenic phenotype following mouse cervical hemisection. They also found that spinal cord astrocytes expressed synapse formation-associated genes to a similar degree across subpopulations, as well as in an unchanged manner between uninjured and SCI conditions. Finally, they confirmed these astrocyte subpopulations are also present in human spinal cord. 

Although it is showing "negative" results, this manuscript makes an important contribution to our understaning of the role that astrocytes play in synaptogenesis after CNS injury. Additionally it is a very well written manuscript with carefully performed state-of-the-art experiments. It should be published in a current form after a spell/editing check. 

Author Response

We thanks Reviewer 1 for the comments on our manuscript. As suggested, we have checked the paper again for spelling/grammar.

Reviewer 2 Report

Spinal cord injury (SCI) is a health issue of world-wide significance, always serious and often fatal. Thus understanding the factors contributing to the inhibition of injury resolution and those promoting healing is of paramount importance. Astrocytes are known to play a pivotal role in these processes and it is anticipated that a better understanding of the molecular mechanisms underlying their function would assist in designing much needed improved treatments.

The work presented in this manuscript by Allahyari and colleagues is an in-depth study on astrocyte heterogeneity following findings by Lin et al. (ref 5), who identified several subpopulations of astrocytes in the mature brain. One such group (population C) demonstrated enhanced synaptogenic characteristics potentially important in the context of spinal cord injury. Allahyari et al. addressed three principal questions emerging from this work in an animal model of SCA: 1) whether the astrocyte population subtype C is overrepresented after injury among the identified populations A-E; 2) whether after SCI, astrocytes respond by inducing the expression of synapse formation-associated genes; and 3) whether the findings in the SCI animal model would be corroborated in human samples. They combined flow cytometry and RNAscope in situ hybridization for the analysis of astrocytes.

Reviewer's critique

The experimental work is well designed and executed to test the hypothesis. The results are clear and their conclusions appropriate. It is an important piece of work that contributes to our understanding on the astrocytic response to spinal cord injury. I support its publication, after a thorough spelling check of the text.

Author Response

We thanks the reviewer for the comments on our manuscript. As suggested, we have again performed a spell-check of the paper.

Reviewer 3 Report

The authors determine the subpopulation of astrocytes type following spinal cord injury using flow cytometry and RNAscope analysis. The study is well designed and written well. It is quite surprising that authors didn’t see many differences in different astrocytes population between the control and SCI animals. Despite the results showed in this study have no differences between control and treated groups. The human data is also a strength for this study. It impacts the role of different astrocytes population in such chronic conditions after injury. Here there are some concerns:

  1. Does author used any positive control to check see if injury occurs as expected in animals (other readouts where authors would expect to see a change).
  2. Does author see any differences in total astrocytes population between control and SCI, as one would expect increased astrocytosis in animals with SCI?
  3. As authors suggest in discussion it would be interesting to check to see if there any significant differences exist at early time points after injury.
  4. At the end of the author’s name “there is an “and” word.
  5. Spelling errors line 49 “multiple”

Author Response

Response to reviewer comments – Allahyari et al – Submission to Cells

Reviewer #2

We thank Reviewer #2 for the insightful comments to improve our study. We have addressed each of the specific comments, and have made changes to the manuscript based on these excellent points.

(1) Does author used any positive control to check see if injury occurs as expected in animals (other readouts where authors would expect to see a change).

Response: In the revised manuscript, we now present quantification of the total number of astrocytes at the C2 lesion location, as an astrogliosis response close to the injury site is - as the reviewer says - a “readout where the authors would expect to see a change” following SCI. Specifically, we quantified the number of Aldh1l1-positive cells in the RNAscope analysis, as Aldh1l1 is a pan-astrocyte marker. At six weeks post-surgery, there was a large increase in total astrocyte numbers in the SCI mice compared to uninjured controls (i.e. approximately a doubling). These data demonstrate that, while we did not observe a change in the distribution of the various astrocyte subpopulations (i.e. Type A-E) after SCI, there was an expected astrogliosis response after SCI that persisted to the long-term six week time point. 

(2) Does author see any differences in total astrocytes population between control and SCI, as one would expect increased astrocytosis in animals with SCI?

Response: As detailed above in response to the reviewer’s first comment, we now present quantification of the total number of astrocytes at the C2 lesion site location. Specifically, we quantified the number of Aldh1l1-positive cells in the RNAscope analysis, as Aldh1l1 is a pan-astrocyte marker. At six weeks post-surgery, there was a large increase in total astrocyte numbers in the SCI mice compared to uninjured controls.

(3) As authors suggest in discussion it would be interesting to check to see if there any significant differences exist at early time points after injury.

Response: We harvested whole-tissue homogenate for the flow cytometry analysis and separate tissue for RNAscope histological analysis only at the single 6 weeks post-injury timepoint; therefore, we don’t have any available samples to perform similar analysis at an earlier time after SCI. We do devote a paragraph in the Discussion section to this important issue of the temporal evolution of the astrocyte response following SCI.

(4) At the end of the author’s name “there is an “and” word.

Response: We have corrected this.

(5) Spelling errors line 49 “multiple”

Response: We have corrected this.